# Deep Eutectic Solvents as a Green Tool for the Extraction of Bioactive Phenolic Compounds from Avocado Peels

**DOI:** 10.3390/molecules27196646

**Published:** 2022-10-06

**Authors:** Beatriz Rodríguez-Martínez, Pedro Ferreira-Santos, Irene Méndez Alfonso, Sidonia Martínez, Zlatina Genisheva, Beatriz Gullón

**Affiliations:** 1Department of Chemical Engineering, Faculty of Science, University of Vigo (Campus Ourense), As Lagoas, 32004 Ourense, Spain; 2Centre of Biological Engineering, Universidade do Minho, 4710-057 Braga, Portugal; 3LABBELS—Associate Laboratory, 4710-057 Braga, Portugal

**Keywords:** green extraction, natural deep eutectic solvents, antioxidant, antibacterial, avocado by-products

## Abstract

Avocado peels are the main agro-industrial residue generated during the avocado processing, being a rich source of bioactive compounds like phenolic compounds. The growing demand for more sustainable processes requires the development of new and effective methods for extracting bioactive compounds from industrial waste. Deep eutectic solvents (DESs) are a new sustainable alternative to toxic organic solvents due to their non-toxicity and biocompatibility. In this study, five selected DESs were applied for the extraction of bioactive phenolic compounds from avocado peels. The extraction efficiency was evaluated by measuring the total phenolics and flavonoids content. The best extraction results were obtained with choline chloride-acetic acid and -lactic acid (92.03 ± 2.11 mg GAE/g DAP in TPC and 186.01 ± 3.27 mg RE/g DAP); however, all tested DESs show better extraction efficiency than ethanol. All the obtained NADES extracts have high antioxidant activity (FRAP: 72.5–121.1 mg TE/g; TAC: 90.0–126.1 mg AAE/g). The synthesized DESs and avocado peels DES extracts had activity against all tested bacteria (*Staphylococcus aureus*, *Streptococcus dysgalactiae*, *Escherichia coli* and *Pseudomonas putida*), and the extracts prepared with choline chloride-acetic acid and -lactic acid have the highest antibacterial activity against all microorganisms. These results, coupled with the non-toxic, biodegradable, low-cost, and environmentally friendly characteristics of DESs, provide strong evidence that DESs represent an effective alternative to organic solvents for the recovery of phenolic bioactive compounds from agro-industrial wastes.

## 1. Introduction

Avocado (*Persea americana* Mill.) belongs to the Lauraceae family and the genus *Persea,* and it is the only species of this genus with economic importance. It is a tropical fruit native to Mexico and Central America [1], but is now grown in many different regions across the globe. Avocado take a significant number of resources to cultivate, since an avocado tree takes between three and five years before it begins bearing fruit. Mexico is the major producer, with more than 234 thousand hectares of avocado trees planted in 2019, up from 168.11 thousand hectares in 2014 [2]. In Europe, Spain is the biggest producer of avocado; 99 thousands of tonnes (95% of all Europe) were produced in 2020 [2]. Fruit is highly appreciated due to its nutrient content. Avocado is known for its health benefits due to the high levels of unsaturated fatty acids, minerals, fiber, proteins, carotenoids, vitamins and polyphenols [1]. The industry only uses the avocado pulp to produce avocado oil and sauces, and produces large quantities of avocado residues, i.e., peel and seeds. These residues account for 25% of the total fruit, and within these the peel itself accounts for between 11% and 17% [1]. Avocado peel is rich in polyphenols, including hydroxycinnamic and hydroxybenzoic acids, flavonoids, flavanols and others [3]. Moreover, numerous biological activities are associated with avocado peel extracts, such as antioxidant, anticancer, antibacterial, anti-inflammatory, anti-allergenic and anti-hypertensive proprieties [1,3].

The extraction of bioactive compounds from avocado peel can be used to further valorize avocado fruit processing in the concepts of biorefinery and the circular economy. Deep eutectic solvents (DESs) were introduced in the beginning of the 21st century, and as a class of a green solvents they have received great attention [4]. DESs have advantages over the traditional solvents as they have a short preparation time, easy storage, low cost, non-flammability, high extraction capacity for both polar and non-polar compounds, and are non-toxic and less harmful for the environment [5]. DESs are based on renewable resources such as carboxylic and amino acids, sugars and amines. The DES based on natural compounds, and metabolites that are naturally present in all types of cells and organisms, are known as natural deep eutectic solvents (NADESs). As NADESs are composed of natural substances, theoretically this makes them fully biodegradable and sustainable [6]. Thanks to the biodegradability, nontoxicity and sustainability, NADES are increasingly attracting interest for applications in several areas such as food, nutraceutical and chemical processes. DESs are a mixture of hydrogen-bond donors (HBD) or hydrogen-bond acceptors (HBAs), at an appropriate molar ratio to form a eutectic mixture. The interaction between HBA and HBD, results in a depression of the melting point of the system relative to its initial components, and is the most important parameter in the formation of these systems [6]. The most used HBA is the choline chloride (ChCl), a quaternary ammonium salt with a melting point of 302 °C; HBDs include organic acids, polyols, sugars and urea.

The use of DESs for the recovery of polyphenolic compounds from a variety of food matrixes has received increasing attention in recent years [7]. DESs are used in extraction processes, food analysis, enzymatic, purification, recycling, fermentation, and other techniques. More specifically, DESs were used to extract hydrophilic and lipophilic compounds of interest [8] from different bioresources, like anthocyanins from jaboticaba (*Myrciaria cauliflora*) fruit waste, tannic acid from onion peels, phlorotannins from algae, phenyletanes and phenylpropanoids from *Rhodiola rosea* L., steroidal saponins from Dioscoreae Nipponicae rhizoma, polyphenols from olive pomace, etc. [9,10,11,12,13,14]. DESs were extensively used to extract polyphenols from plants such as: olive, mate, moringa and mulberry leaves [15,16,17,18]. It was concluded that in addition to their non-toxic and environmentally friendly nature, the use of DESs improved the polyphenol extraction from the plant materials, compared to the commonly used organic solvents [15,18].

As far as we know, DESs have never been used before for the extraction of polyphenol compounds from avocado peel. In this context, the main objective of this study was to add value to avocado peels, as a by-product of the food industry, for the recovery of bioactive compounds through the green extraction with alternative solvents—DESs. The work consists of the synthesis and selection of DESs for an efficient extraction of phenolic compounds, and the characterization of the extracts in terms of chemical—Total Phenolic Content (TPC) and Total Flavonoid Content (TFC) using colorimetric methods, and phenolic profile using liquid chromatography (UHPLC)—and bioactive properties—antioxidants and antibacterial.

## 2. Results and Discussion

### 2.1. Screening of the Solvent for Phenolic Compounds Recovery

The choice of the suitable DES for phenolic compound extraction was highly dependent on some factors like the composition of the raw material, the diffusion rate of the target compounds in the solvent, surface tension, the viscosity of the DES, and time extraction process [5]. In this sense, the increase in extraction temperature can lead to reduced viscosity of the DES, which consequently results in a decrease in surface tension and an increase in molecules diffusion, improving the extraction of phenolic compounds.

Most of the DESs proposed so far are based on renewable sources, such as organic acids, amino acids, sugars and amines, representing a new generation of green solvents [19]. As mentioned before, these DESs based on natural compounds (NADES) mostly represent a class of low toxicity and biodegradable solvents. Depending on the formulation, some NADES can dissolve chemicals (natural or synthetic) with different polarities. In addition, as they are mostly composed of natural substances, they can be a greener alternative for applications that usually involve some organic solvents and ionic liquids [6].

Recently, natural DES have been widely applied to the efficient recovery (and green alternative) of phenolic compounds from various industrial food and agroforestry biowastes [5,7,20,21,22,23]. Nevertheless, no application on the recovery of bioactive molecules from avocado by-products has been found in the literature. Therefore, this study evaluated the possibility of DES in the recovery of bioactive phenolic compounds from avocado peel.

Figure 1 shows the TPC and TFC values obtained for the biocompound extractions carried out with five tested DESs in dried avocado peels (DAP). To compare the extraction efficiencies of the DES with those obtained from traditional organic solvents, extractions with a mixture of ethanol 96% (*v*/*v*) were performed in parallel under the same operational conditions. It is observed that the DES extracts present higher phenolic content than the control solvent ethanol 96% (25.27 ± 0.92 mg GAE/g DAP). The highest TPCs provided by DES correspond to DES 2 (92.03 ± 2.11 mg GAE/g DAP), composed by acetic acid, ChCl and water, and DES 5 (92.09 ± 4.92 mg GAE/g DAP), composed by lactic acid and ChCl. On the other hand, when flavonoid content was evaluated, the maximum TFC values obtained for DES coincided wide DES 5 (211.73 ± 1.50 RE/g DAP) and DES 2 (186.01 ± 3.27 RE/g DAP), which have a 50% higher content of flavonoids than ethanol (94.70 ± 3.86 mg RE/g DAP). In our previous work with avocado peel [3], the maximum phenolic/flavonoid contents recovered by conventional solvents (water and ethanol) under optimal conditions (ethanol concentration of 38.46% (*v*/*v*) and 44.06 min) were TPC = 45.34 ± 1.7 mg GAE/g DAP and TFC = 87.56 ± 1.2 mg RE/g DAP. These values are significantly lower than those obtained for extraction performed by all the DES used in this work.

Several studies have reported the performance of DES in the extraction of phenolic compounds from agri-food residues. For example, the results of Peng et al. [24] indicated that some DESs showed remarkable effects on the extraction efficiency of phenolic acids from *Lonicerae japonicae* Flos. In another study, Oliveira et al. [25] used various DES-based menthol and ChCl to extract bioactive compounds from *Curcuma longa* L., and their results showed that DES composed of menthol + acetic acid (1:1), ChCl + lactic acid (1:2) and ChCl + acetic acid (1:2) show great capacity to recover bioactive flavonoids from the flowers, leaves and rhizome of *Curcuma longa*. Furthermore, and similar to our work, the authors showed that these DES are more efficient than ethanol in recovering phenolic compounds.

Therefore, in the case of avocado peels it can be concluded that DES 5 is the most suitable solvent to be used in the extraction of antioxidants in terms of phenolics and flavonoids content. However, the close values of TPC and TFC reported by DES 2 together with a lower cost in its synthesis, 8.75 €/kg, versus 33.50 €/kg for DES 5 (Table 1), allow us to conclude that the optimal DES set could be the DES 2.

### 2.2. Phenolic Compound Identification and Quantification

Phenolic compounds are complex molecules, and their extraction from a solid biomass requires compatible solvents. Identification and quantification of individual compounds present in the obtained DES extract was performed. The methodology and technology used permitted the identification of seven phenolic compounds (Table 2). These compounds were identified by comparison of their UV spectra and retention times with those obtained in previous findings, as well as with the commercial standards (chromatogram example in Appendix A).

Catechin was the phenol with the highest concentrations in all extracts. This is in accordance with previous studies of the phenolic composition of avocado peel, and where catechin was also found to be the most abundant phenol compound [29]. Between extracts, the highest concentrations of catechin were registered in the extracts made with DES 2 (ChCl: acetic acid: water (1:1:10)), representing 76% of the total concentration of phenolic compounds. In all extracts, catechin represents between 65% and 76% of identified phenolic compounds. Besides catechin, the compounds rutin and epicatechin were previously found in extracts of avocado peel [3,30]. In the present samples, rutin was the second most abundant phenolic compound in all extracts. Rutin was found at concentrations between 82 and 103 mg/100 g in all samples except in the ethanol extracts, where it presented the lowest values (44 mg/100 g). Epicatechin and gallic aid were also found in all extracts. The highest concentrations of gallic acid were registered in extracts made with DES 1 (76.7 ± 0.5 mg/100 g) and DES 5 (71.0 ± 1.3 mg/100 g). Whereas the highest concentrations of epicatechin were found in extracts made with DES 2 (35.3 mg/100 g), DES 4 (33.3 mg/100 g) and ethanol (32.8 mg/100 g), the acids 3,4 hydroxybenzoic and 2,5 hydroxybenzoic were only found in some samples. The extracts made with DES 1 (lactic acid: sodium acetate (3:1)) did not have any of these hydroxybenzoic acids. Ferulic acid was identified in all samples in concentrations close to 6–8 mg/100 g, except in the ethanol extracts, where it presented the lowest values (3.3 mg/100 g). Extracts made with DES 2 had the highest total concentration of individual phenolic compounds (829 mg/100 g), followed by extracts made with DES 5 (730 mg/100 g) and DES 1 (686 mg/100 g). This is to be expected as the total concentration of individual phenolic compounds is mainly influenced by the concentration of catechin. Extracts with the highest concentrations of catechin accounted for the highest total concentrations and so on. These results are in accordance with the results of TPC and TFC (Figure 1), where DES 2, followed by DES 5 extracts, accounted for the highest concentration of TPC and TFC. As a final remark, it is worth mentioning that this is the first study of the individual phenolic compounds of extracts from avocado peel made with DESs.

### 2.3. Antioxidant Activity of Extracts

The antioxidant capacity of biocompounds is one of the most studied actions, referenced as a mechanism to prevent oxidative stress associated with several diseases, as well as in food preservation and enrichments. Moreover, the antioxidant activity of the natural extracts (from plants, agri-food waste, etc.) depends on the composition and structure of the biocompounds, such as phenolic acids and flavonoids and their ability to neutralize free radicals, i.e., chelators and free radical scavengers’ activities [31].

In the food context, the emergence of new antioxidant sources has been the target of recent research and applications in food protection and nutritional enrichment, promoting the consumption of bioactive compounds.

In this work, besides the evaluation of the potential of DESs as extractants of poorly water-soluble compounds, we also studied whether their presence in the final product could potentiate the bioactivity of the extracts. In that way, antioxidant activity was evaluated.

To assess the impact of the DES and ethanol extraction on the antioxidant potential of the extracts obtained from avocado peel, ferric reducing antioxidant power (FRAP) and TAC assays were performed. The FRAP method is based on the ability of samples to donate electrons to reduce a Fe^3+^ -TPTZ complex to a blue Fe^2+^ -TPTZ complex, and TAC is based on formation of the green-colored complex of phosphate/Mo(V) at acidic pH values. The results from these two tests are shown in Table 3.

Our data show that all extracts obtained from avocado peel, regardless of the extraction solvent, have considerable antioxidant activity. Moreover, pure DES showed no antioxidant activity for the methods tested. The FRAP assay proved the potent reducing power of extracts obtained with DES, showing higher values for extracts obtained with DES 5, 2 and 1 (121.09 ± 8.9, 115.4 ± 7.5, and 107.32 ± 6.4 mg TE/g DAP, respectively). In our previous work with avocado peel, the optimized extract showed that in the FRAP assay the hydroethanolic extracts had values of 44.65 mg TE/g DAP [3]. This value is like the value obtained for the ethanol extract obtained in this work (46.37 ± 3.4 mg TE/g DAP), and lower than all extracts obtained with DES.

Like the results of FRAP assay, the highest TAC values were reported in the DES 5, DES 1 and DES 2 (126.07 ± 11.5, 122.41 ± 11.8 and 121.85 ± 7.6 mg AAE/g, respectively). In addition, ethanol showed lower values (61.88 ± 7.2 mg AAE/g DAP) than all extracts obtained with the tested DES.

Suleria et al. [32] performed the screening and characterization of phenolic compounds and studied the antioxidant capacity of different fruit peels, reporting that avocado peel presents values of 3.65 ± 0.07 mg AAE/g DAP for the FRAP method and 4.50 ± 0.16 mg AAE/g DAP for TAC. These values are much lower than those obtained in this work. On the other hand, Dibacto et al. [33] reported TAC values for avocado peel extracts of 129.47 mg AAE/g DAP using water as solvent, 132.87 mg AAE/g DAP with 50% ethanol, and 130.02 mg AAE/g DAP for pure ethanol. These results are more like those obtained in the current investigation. The differences in the results may be due to several factors, such as the origin of the raw material, drying and storage conditions, extraction methodology and quantification techniques. On the other hand, we must consider the possibility of degradation of phenolic compounds under adverse environmental conditions, which can affect their antioxidant activity. The results of antioxidant activity are in accordance with the results of TPC, TFC and individual phenolic compounds. In other words, the fact that the extractions with the most antioxidant activity are those that have the highest content of phenolic and flavonoids compounds allows us to conclude that it is these compounds that provide the antioxidant capacity to the sample (see Pearson’s correlation in Table 4 (general correlation), and Appendix A (different extracts correlation).

Several studies have been carried out to evaluate the antioxidant activity (and other bioactivities) of some phenolic compounds. Grzesik et al. [34] described how the excellent antioxidant properties of catechins and other flavonoids (including rutin), as well as phenolic acids, make them ideal candidates for food (preventing the oxidation and increasing its antioxidant content) and therapeutic (attenuating oxidative stress and regulating body homeostasis) applications.

In our case, we must consider the individual effect of each compound present in avocado peel extracts (rich in catechins, rutin and phenolic acids), but also their synergistic effect that may enhance (or not) the antioxidant action of the extracts.

### 2.4. Antibacterial Activity of Extracts

Determining the ability of biocompounds/extracts to inhibit or reduce microbiological growth is highly valued by the industry. The application of these extracts as an additive in foods, in addition to increasing the content of antioxidants ingested by the consumer, also potentiates food preservation. Table 5 presents the results of antibacterial activity for the extracts obtained from avocado peels, and the pure DES activity. The agar well diffusion method is a suitable visual method to evaluate bacteria inhibition [35]. Four bacteria of interest in the food industry have been tested, suggesting probable applications of the obtained extracts in food preservation and/or for coating films. Two were for Gram-positive bacteria (*Staphylococcus aureus* and *Streptococcus dysgalactiae* subsp. *equisimilis*) and two for Gram-negative (*Escherichia coli* and *Pseudomonas putida*). From the 5 DES synthetized in this work, all of them showed high antimicrobial activity, except DES 3 for *E. coli*, which seems to be resistant because it does not have an inhibition halo, and for *P. putida* which has a halo of 12 mm (Table 5). This fact is quite normal due to the circumstance that Gram-negative bacteria present an external membrane in their structure.

An increase in studies on the antimicrobial activity of several DESs has been observed in the literature [36,37,38,39]. Although organic acids are associated with the antibacterial activity found for these specific DES, the other compounds also appear to be responsible [37,40]. Hong et al. [41] stated that some organic acids, like acetic, lactic and citric, are more effective against *E. coli* than *S. aureus*. In our case, only DES 5 showed greater activity for *E. coli* compared to *S. aureus*.

To our knowledge, there are no extracts obtained with DES from avocado peels. Therefore, this is the first study to demonstrate the antimicrobial activity of these extracts. All extracts obtained from DAP showed higher activity compared to the respective pure solvent, except the extract obtained with DES 3, and the ethanolic one in the case of *E. coli*.

As we can see in Table 5, the extracts obtained with DES 2 (ChCl-acetic acid) and DES 5 (ChCl-lactic acid) are the ones with the highest antimicrobial activity for all the bacteria used in this work. Furthermore, only the extract obtained by DES 3 appears to have no activity against these microorganisms. Ethanol extracts have some antimicrobial activity, especially against Gram-positive bacteria (intermediate action with an inhibition halo of 14 mm for *S.aureus* and 15.5 for *S.equisimilis*).

These results can be explained by the content of phenolic compounds in the extracts. As we can see in Figure 1 and Table 2, the extracts obtained with DES 2 and DES 5 are those with the highest content of bioactive compounds (TPC and TFC), and therefore, they are correlated with high antioxidant and antibacterial activity.

Recently, Ivanovic et al. [39] studied the extraction process of phenolic compounds recovery from *Achillea millefolium* L. with different solvents (MeOH, EtOH, and five NADES based on ChCl as a hydrogen bond acceptor (HBA) and different hydrogen bond donors (HBD: lactic acid, 1,4-butanediol, 1,2-propanediol, fructose and urea)). They showed that extracts obtained with NADES have high antimicrobial activity compared to extracts obtained with organic solvents. These results, similar to ours, lead to affirm the potential of DES as alternative solvents for the extraction of bioactive compounds, providing and increasing the activity of the obtained extract.

## 3. Materials and Methods

### 3.1. Chemicals and Plant Material

Avocado (*P. americana*, Mill.) peels from the Hass variety were collected from local restaurants in Ourense (Spain). The raw material was washed with tap water to remove the remains of pulp and dried in an oven (JP Selecta Theroven) at 50 °C for 24 h until they achieved a constant humidity of around 6.5%. Subsequently, the DAP were ground to a particle size between 0.3–1 mm using a Polymix PX-MFC 90D and sieved. DAP was stored in plastic bags at a temperature of −20 °C until use.

Acetic acid (96%), ammonium molybdate, ascorbic acid, choline chloride, citric acid, ethanol, Folin–Ciocalteu reagent, gallic acid, glycerol, hydrochloric acid, iron(III) chloride hexahydrate, lactic acid (90%), methanol, rutin, TPTZ (2,4,6-tri(2-pyridyl)-S-triazine), trolox (6-hydroxy-2,5,7,8-tetramethylchroman-2-carboxylic acid), sodium acetate, sodium carbonate, sodium phosphate and sulphuric acid were obtained from Sigma-Aldrich (Barcelona, Spain).

### 3.2. Deep Eutectic Solvents Preparation

DES were synthesized based on the methodology described by García et al. [43] with minor modifications. Briefly, the procedure consists of mixing a hydrogen bond acceptor (e.g., choline chloride) with a hydrogen bond donor (carboxylic acid, alcohols, etc.) in a capped bottle. This concoction is heated for 30–120 min at a temperature of 80 °C until it adopts a colorless appearance. Table 1 shows the different DES employed in the current work with their abbreviations and the molar ratio of their components. These DES were selected based on recent literature with other raw materials [36,37,38]. Except DES 2, which already contains water in its formulation, 30% water was incorporated in all of them to reduce viscosity and facilitate polyphenols diffusion [26].

### 3.3. Solid–Liquid Extraction of Phenolic Compounds from Avocado Peel

For the extraction of phenolic compounds, five different DES were selected. Ethanol 96% (*v*/*v*) was used as the control extraction solvent. The extraction process was performed in an orbital shaker (Adolf Kühner AG, Birsfelden, Switzerland) using Erlenmeyer flasks (duly protected from the light with aluminum foil) with a capacity of 100 mL and working with a 1:15 (*w*/*w*) sample to solvent ratio. The extraction was carried out at 50 °C for 120 min at a shaking speed of 150 rpm. The conditions were selected based on previous experiments and other research with different raw materials [44,45]. After extraction, the supernatant was separated from the solids by filtration through filter paper (Whatman Ashless, Grade 42) and stored at −20 °C until further use. All extractions were performed in triplicate.

### 3.4. Determination of Total Phenolic and Flavonoid Content

The TPC was determined by the Folin–Ciocalteu method [3] with some minor modifications. Shortly, 0.3 mL of diluted extracts were mixed with 2.5 mL of a diluted Folin–Ciocalteu reagent (10 times) and 2 mL of Na_2_CO_3_ solution (7.5% *w*/*v*). After vigorous shaking, samples were incubated at room temperature for 1 h and their absorbance was measured at 760 nm. Gallic acid was used as a standard and the results were expressed in milligrams of gallic acid equivalent (GAE) per gram of DAP.

The TFC was determined using the method described by Blasa et al. [46], which consists in mixing 1 mL of diluted extracts with 0.3 mL of 5% NaNO_2_. After 5 min, 0.3 mL of AlCl_3_ (10%) are added. It is incubated for 6 min and 2.0 mL of NaOH (1 M) was incorporated. Vigorous agitation is performed, and after 5 min of incubation the absorbance was measured at a wavelength of 510 nm. Rutin was used as a standard, expressing the results in milligrams of rutin equivalents (RE) per gram of DAP.

### 3.5. Identification and Quantification of Individual Phenolic Compounds

DESs and extracts of avocado peel were analysed using a Shimatzu Nexpera X2 UPLC chromatograph equipped with a Diode Array Detector (DAD) (Shimadzu, SPD-M20A, Kyoto, Japan), following the method described by Ferreira-Santos et al. [47]. Separation was executed at 40 °C, with a flow rate of 0.4 mL/min on a C18 reversed-phase column by Waters (Acquity UPLC^®^ BEH column, 2.1 mm × 100 mm, 1.7 μm particle size) and a pre-column of the same material. HPLC grade solvents water/formic acid 0.1% (A) and acetonitrile (B) were used. The elution gradient for solvent B was as follows: from 0.0 to 5.5 min eluent B at 5%, from 5.5 to 17 min linearly increasing from 5 to 60%, from 17.0 to 18.5 min a linearly increasing from 60 to 100%; finally, the column was equilibrated at 5% for 11.5 min. The identification of the phenolic compounds was made by comparing their UV spectra and retention times with that of corresponding standards. Quantification was carried out using calibration curves for each phenolic compound using concentrations between 250 and 2.5 mg/L, and the limit of detection (LOD) and limit of quantification (LOQ) were calculated (data not shown). In all compounds, the coefficient of linear correlation (R^2^) was higher than 0.990. Different wavelengths (209–370 nm) were used for the quantification and identification of the target compounds. The values of individual phenolic compounds were expressed in milligrams per 100 g of avocado peel (mg/100 g). All analyses were made in triplicate. All standards were of analytical grade and procured from Sigma Aldrich (St. Louis, MO, USA).

### 3.6. Antioxidant Activity Evaluation

To determine the antioxidant activity, FRAP and TAC methods were performed.

The FRAP assay was performed following the methodology described by Gullón et al. [48]. The FRAP reagent was prepared by mixing 50 mL of acetate buffer (300 mM, pH 3.6) with 5 mL of TPTZ solution (10 mM) and 5 mL of FeCl_3_·6H_2_O solution (20 mM). Then, 0.1 mL of the extracts will be mixed with the prepared FRAP reagent and, after 6 min of incubation, the absorbance at 593 nm is measured. Trolox was used as a standard, expressing the results in milligrams of trolox equivalents (TE) per gram of DAP.

The TAC assay was carried out using the phospho-molybdenum method described by Prieto et al. [49] with minor modifications. Shortly, 2 mL of a reagent solution (0.6 mol/L sulfuric acid, 28 mmol/L sodium phosphate and 4 mmol/L ammonium molybdate) were mixed with 0.2 mL of the sample. The mixture is incubated at 95 °C for 90 min and, when cooled to room temperature, its absorbance at 695 nm is measured against a blank solution. Ascorbic acid was used as a standard, expressing the results in milligrams of ascorbic acid equivalents (AAE) per gram of DAP.

### 3.7. Antimicrobial Activity Evaluation

The antibacterial activity was evaluated for all synthetized DESs and for the produced avocado peel extracts. Two Gram-negative bacteria, *Escherichia coli* (ATCC 25922) and *Pseudomonas putida* (ATCC 700801), and two Gram-positive bacteria, *Streptococcus dysgalactiae* subsp. *equisimilis* (CECT 926) and *Staphylococcus aureus* (ATCC 6538) obtained from the Centre of Biological Engineering stock collection (University of Minho, Braga, Portugal), were used. Bacteria strains were replicated on blood agar and incubated at 37 °C for 24 h to ensure that bacterial cells were in the exponential growth phase. Single colonies were inoculated using sterile water and the bacterial suspensions were adjusted to a concentration of ~1.5 × 10^8^ CFU/mL (0.5 McFarland). Agar well diffusion method was used to screen the antibacterial activity of different DESs/extracts as demonstrated by Gonelimali et al. [50]. Twenty mL of Nutrient Agar culture medium was then poured into the Petri dish. Upon solidification, 1 mL of fresh bacterial culture was pipetted and spread on the surface of the agar. After that, wells were made using a sterile cork borer (6 mm in diameter) into agar plates containing inoculums. Then, 100 μL of each DESs/extract was added to respective wells. The plates were incubated at 37 °C for 24 h. Antibacterial activity was assessed by measuring the zone of inhibition (including the wells diameter) after the incubation period. The experiments were performed in triplicate.

### 3.8. Statistical Analysis

The data were examined by analysis of variance (ANOVA). The least significant difference (LSD) test was applied, with a 95% confidence interval (*p* ≤ 0.05), for comparison of the mean values by using the statistical software Statistica version 8.1 (Statsoft© Inc., Tulsa, OK, USA). Pearson’s correlation coefficients were used to establish the relationship between the content of TPC and TFC and antioxidant capacity. Multiple regression and multivariate data analysis such as the partial least squares coefficient method were carried out.

## 4. Conclusions

In this study, a low-cost, more environmentally friendly and effective extraction method (based on NADES) was applied for the recovery of bioactive phenolic compounds from avocado peels. Based on the experiential results, it can be concluded that acetic acid and lactic acid-based DES (ChCl-acetic acid and ChCl-lactic acid) are promising solvents, with extraction results higher than those obtained with the other DESs and ethanol. Moreover, the antioxidant capacity of the DES extracts was higher in all cases compared to ethanolic extracts, due to the greater number of phenolic compounds. The synthetized DES and extracts obtained with DES had great activity against all tested bacteria, and on the other hand, the extracts prepared with ChCl-acetic acid and ChCl-lactic acid have the highest antibacterial activity against all microorganisms. These results, coupled with the non-toxic, biodegradable, low-cost, and environmentally friendly characteristics of DESs, provide strong evidence that DESs represent a better alternative to organic solvents for the recovery of phenolic bioactive compounds from agro-industrial wastes.

## Figures and Tables

**Figure 1 molecules-27-06646-f001:**
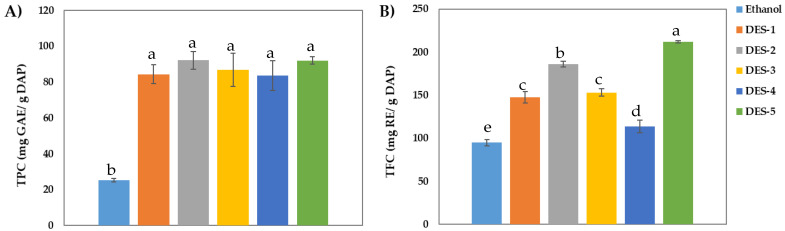
(**A**) Total phenolic content (TPC) and (**B**) total flavonoid content (TFC) obtained for the avocado peel extractions carried out with DES and ethanol. Data are means ± standard deviation (*n* = 3). Different letters indicate a statistical difference (*p* < 0.05).

**Table 1 molecules-27-06646-t001:** Components, molar ratio, price and abbreviations of deep eutectic solvents (DES) employed in this work.

Abbreviations	Component 1	Component 2	Component 3	Molar Ratio	Price (€/kg)	References
DES 1	Lactic acid	Sodium acetate	-	3:1	119.91	El Kantar et al. [26]
DES 2	Acetic acid	Choline chloride	Water	1:1:10	8.75	Hernández-Corroto et al. [27]
DES 3	Glycerol	Choline chloride	-	3:1	129.93	Ozturk et al. [7]
DES 4	Glycerol	Citric acid	-	2:1	96.03	Bajkacz and Adamek [28]
DES 5	Lactic acid	Choline chloride	-	3:1	33.50	El Kantar et al. [26]

**Table 2 molecules-27-06646-t002:** Individual phenolic compounds identified and quantified in the extracts from avocado peel obtained with DES and ethanol.

Phenolic Compound	Catechin (mg/100 g)	3,4 HBA (mg/100 g)	2,5 HBA (mg/100 g)	Gallic acid (mg/100 g)	Epicatechin (mg/100 g)	Ferulic acid (mg/100 g)	Rutin (mg/100 g)	Total (mg/100 g)
Ethanol	236.4 ± 10.3 ^e^	0.3 ± 0.0 ^c^	n.d.	0.9 ± 0.1 ^e^	32.8 ± 0.3 ^b^	3.3 ± 0.0 ^e^	44.0 ± 0.2 ^e^	318
DES 1	478.4 ± 2.4 ^bc^	n.d.	n.d.	76.7 ± 0.5 ^a^	25.9 ± 0.1 ^c^	6.8 ± 0.1 ^c^	98.0 ± 0.5 ^b^	686
DES 2	319.0 ± 7.1 ^a^	17.1 ± 2.6 ^a^	12.1 ± 0.1 ^b^	22.3 ± 2.3 ^d^	35.3 ± 0.3 ^a^	7.6 ± 0.0 ^b^	103.1 ± 0.2 ^a^	829
DES 3	406.7 ± 46.5 ^d^	8.4 ± 0.0 ^b^	n.d.	1.7 ± 0.1 ^e^	17.8 ± 0.1 ^d^	8.0 ± 0.0 ^a^	95.5 ± 1.0 ^c^	450
DES 4	521.5 ± 73.5 ^c^	15.9 ± 0.7 ^a^	22.0 ± 1.2 ^a^	55.0 ± 0.7 ^c^	33.3 ± 2.7 ^ab^	6.1 ± 0.1 ^d^	81.9 ± 0.3 ^d^	621
DES 5	347.9 ± 49.0 ^b^	n.d.	10.9 ± 0.2 ^c^	71.0 ± 1.3 ^b^	18.1 ± 0.1 ^d^	6.4 ± 0.1 ^d^	102.2 ± 0.1 ^a^	730

Data are means ± standard deviation (*n* = 3). HBA: hydroxybenzoic acid. Values in the same column followed by different letters are significantly different (*p* < 0.05). n.d.: No detected.

**Table 3 molecules-27-06646-t003:** Antioxidant capacity of the phenolic extracts from avocado peel obtained with five DES and ethanol.

Extract	FRAP (mg TE/g DAP)	TAC (mg AAE/g DAP)
Ethanol	46.4 ± 3.4 ^d^	61.9 ± 7.2 ^c^
DES 1	107.3 ± 6.4 ^b^	122.4 ± 11.8 ^a^
DES 2	115.4 ± 7.5 ^ab^	121.9 ± 7.6 ^a^
DES 3	84.5 ± 2.7 ^c^	90.0 ± 7.7 ^b^
DES 4	72.5 ± 3.5 ^c^	91.1 ± 3.8 ^b^
DES 5	121.1 ± 8.9 ^a^	126.1 ± 11.5 ^a^

Results were expressed in mean ± standard deviation (*n* = 3). Values in the same column followed by different letters are significantly different (*p* < 0.05).

**Table 4 molecules-27-06646-t004:** Pearson’s correlation coefficients for the total phenolic content (TPC), total flavonoid content (TFC) and antioxidant capacity (FRAP and TAC) of extracts from avocado peel. Significant correlations are marked in bold.

Variables	TPC	TFC	FRAP	TAC
TPC	1.00	-	-	-
TFC	**0.73**	1.00	-	-
FRAP	0.83	**0.93**	1.00	-
TAC	**0.82**	**0.85**	**0.98**	1.00

**Table 5 molecules-27-06646-t005:** The antibacterial capacity of phenolic extracts from avocado peel and pure DES.

Bacteria	Gram-Positive				Gram-Negative			
	*Staphylococcus aureus*	*R/I/S*	*Streptococcus dysgalactiae* subsp. *equisimilis*	*R/I/S*	*Escherichia coli*	*R/I/S*	*Pseudomonas putida*	*R/I/S*
**Antibiotic (mm)**	34.0 ± 1.0 ^c^ (AMP)	*S*	31.0 ± 0.0 ^d^ (AMP)	*S*	27.0 ± 2.0 ^ef^ (AMP)	*S*	26.0 ± 1.0 ^g^ (TC)	*S*
**Pure DES (mm)**								
DES 1	33.0 ± 0.0 ^c^	*S*	38.0 ± 1.0 ^b^	*S*	31.5 ± 0.5 ^d^	*S*	36.0 ± 0.5 ^cd^	*S*
DES 2	29.5 ± 0.5 ^d^	*S*	37.5 ± 0.5 ^b^	*S*	23.5 ± 0.5 ^f^	*S*	34.0 ± 1.0 ^de^	*S*
DES 3	15.0 ± 1.0 ^e^	*I*	n.a.	*I*	n.a.	*R*	12.0 ± 0.5 ^h^	*R*
DES 4	27.0 ± 1.5 ^d^	*S*	29.5 ± 0.5 ^d^	*S*	24.0 ± 1.0 ^f^	*S*	29.5 ± 1.5 ^f^	*S*
DES 5	27.5 ± 1.5 ^d^	*S*	33.5 ± 0.5 ^c^	*S*	29.5 ± 0.5 ^e^	*S*	35.5 ± 0.5 ^cd^	*S*
DMSO 10%	n.a.	*R*	n.a.	*R*	n.a.	*R*	n.a.	*R*
**Extracts (mm)**								
DES 1	40.0 ± 1.0 ^b^	*S*	42.5 ± 0.5 ^a^	*S*	36.0 ± 0.5 ^b^	*S*	43.5 ± 1.5 ^a^	*S*
DES 2	42.0 ± 0.0 ^a^	*S*	42.0 ± 1.0 ^a^	*S*	39.0 ± 1.0 ^a^	*S*	40.0 ± 1.0 ^b^	*S*
DES 3	16.0 ± 1.0 ^e^	*I*	17.0 ± 1.0 ^e^	*I*	n.a.	*R*	14.0 ± 1.0 ^h^	*I*
DES 4	30.5 ± 0.5 ^d^	*S*	32.5 ± 0.5 ^c^	*S*	28.5 ± 0.5 ^e^	*S*	33.5 ± 0.5 ^e^	*S*
DES 5	33.5 ± 0.5 ^c^	*S*	38.0 ± 1.0 ^b^	*S*	34.5 ± 0.5 ^c^	*S*	38.5 ± 1.5 ^b^	*S*
Ethanol	14.0 ± 1.0 ^e^	*I*	15.5 ± 0.5 ^e^	*I*	n.a.	*R*	10.5 ± 2.0 ^h^	*R*

Results were expressed in mean ± standard deviation. *S* = susceptible ≥ 18 mm. *I* = intermediate 13 to 17 mm. *R* = resistant ≤ 12 mm. Source: [42]. Values in the same column followed by different letters are significantly different (*p* < 0.05). AMP: ampicillin (1 mg/mL); TC; tetracycline (2 mg/mL); n.a.: No activity.

## Data Availability

Not applicable.

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
