# Peer review of "Deep Eutectic Solvents as a Green Tool for the Extraction of Bioactive Phenolic Compounds from Avocado Peels"

_molecules, 2022, doi:10.3390/molecules27196646_

Round 1
Reviewer 1 Report
The paper titled "Deep Eutectic Solvents as a Green Tool for the Extraction of Bioactive Phenolic Compounds from Avocado Peels" represents one of the many scientific papers that have appeared recently related to DES and agroindustrial waste materials. However, as the authors mentioned, there is no work on the application of NADES for the extraction of phenolic compounds from avocado peels. Therefore, this work presents a way to use both waste materials and non-toxic solvents. The authors have prepared the manuscript correctly and some comments are given below.
Comments to the authorsLines 85 - 88: It looks like that total phenols and total phavonoids were determined by HPLC. Please rewrite these sentences and make a correct distinction between TP, TF and phenolic compounds.
Line 114: 96%-? Please delete - after 96%.
Lines 118 and 119: Please check the mode of the units writing for the determination of flavonoids. Please provide the correct units. See also phenols.
Please explain why flavonoid levels in some extracts are higher than phenols. See also Fig. 1
Line 140: Please delete concentration and put the terms content.
Why did the authors report the concentration of phenolic compounds in mg/L and the total phenolic and flavonoid content in mg/g? Please indicate the uniformity.
Table 1. Please provide the units for phenolic compounds in the table. See table 2.
Table 3. please indicate the units in the table, not below the table.
Table 2. five DES instead of 5
Lines 235 - 236: Please explain this observation.
Line 336: Did the authors use diluted or undiluted extracts for the determination of flavonoids?
Author Response
We would like to thank the editor and reviewers that spent time in this process and add value to the manuscript. The manuscript has been improved according to the suggestions of reviewers; The alterations are represented in the manuscript in red.
The paper titled "Deep Eutectic Solvents as a Green Tool for the Extraction of Bioactive Phenolic Compounds from Avocado Peels" represents one of the many scientific papers that have appeared recently related to DES and agroindustrial waste materials. However, as the authors mentioned, there is no work on the application of NADES for the extraction of phenolic compounds from avocado peels. Therefore, this work presents a way to use both waste materials and non-toxic solvents. The authors have prepared the manuscript correctly and some comments are given below.
Reply: We would like to thank the reviewer for their comments and relevant suggestions, which have greatly improved the quality of the manuscript.
Comments to the authors:
- Lines 85 - 88: It looks like that total phenols and total flavonoids were determined by HPLC. Please rewrite these sentences and make a correct distinction between TP, TF and phenolic compounds.
Reply: Done. The sentences have been rewritten according to the recommendation (line 93-95)
- Line 114: 96%-? Please delete - after 96%.
Reply: Done. The “-“ was deleted.
- Lines 118 and 119: Please check the mode of the units writing for the determination of flavonoids. Please provide the correct units. See also phenols.
Reply: The mode of the units is correct. In TPC, gallic acid is used as the standard so the units are expressed in gallic acid equivalents (GAE); while in TFC, the standard used is rutin, so the units are expressed in rutin equivalents (RE) per gram of raw material. This standard was used in previous studies for other raw materials.
Kaundal, R., Kumar, M., Kumar, S., Singh, D., Kumar, D., 2022. Polyphenolic Profiling, Antioxidant, and Antimicrobial Activities Revealed the Quality and Adaptive Behavior of Viola Species, a Dietary Spice in the Himalayas. Molecules 27, 1-19. https://doi.org/ 10.3390/molecules27123867
Stanius, Ž., Dūdėnas, M., Kaškonienė, V., Stankevičius, M., Skrzydlewska, E., Drevinskas, T., Ragažinskienė, O., Obelevičius, K., Maruška, A., 2022. Analysis of the Leaves and Cones of Lithuanian Hops (Humulus lupulus L.) Varieties by Chromatographic and Spectrophotometric Methods. Molecules 27, 1–13. https://doi.org/10.3390/molecules27092705
Zhang, K., Han, M., Zhao, X., Chen, X., Wang, H., Ni, J., Zhang, Y., 2022. Hypoglycemic and Antioxidant Properties of Extracts and Fractions from Polygoni Avicularis Herba. Molecules 27, 1–16. https://doi.org/10.3390/molecules27113381
- Please explain why flavonoid levels in some extracts are higher than phenols. See also Fig. 1
Reply: Due to the units of measurement are not the same (flavonoids are measured in rutin equivalents and phenols in gallic acid equivalents), the values obtained cannot be compared with each other.
- Line 140: Please delete concentration and put the terms content.
Reply: Done.
- Why did the authors report the concentration of phenolic compounds in mg/L and the total phenolic and flavonoid content in mg/g? Please indicate the uniformity.
Reply: Concentration units were unified throughout the manuscript. Table 1 where the individual phenolic compounds are represented, as well as the TPC and TFC were expressed in relation to the mass of avocado peel used in the extraction (mg/g or mg/100g).
- Table 1. Please provide the units for phenolic compounds in the table. See table 2.
Reply: Done.
- Table 3. please indicate the units in the table, not below the table.
Reply: Done.
- Table 2. five DES instead of 5
Reply: Done.
- Lines 235 - 236: Please explain this observation.
Reply: An explanation was included in the manuscript to clarify the observation. (Line 247-256)
- Line 336: Did the authors use diluted or undiluted extracts for the determination of flavonoids?
Reply: In the determination of flavonoids, diluted extracts were used. A clarification was added in the manuscript text. (line 356)
Reviewer 2 Report
In this study, five selected DESs were applied for the extraction of bioactive phenolic compounds from avocado peels. The extraction efficiency was evaluated by measuring the total phenolics and flavonoids content. All the results coupled with the non-toxic, biode-gradable, low-cost, and environmentally friendly characteristics of DESs, provide strong evidence that DESs represent an effective alternative to organic solvents for the recovery of phenolic bioactive compounds from agro-industrial wastes. This study is very meaningful to recycle the by-product in food industry. However, there are some critical issues should be clarified as follows:
1. The total phenolic compounds contents data should contain standard deviation data in table 1.
2. For the data in Table 1 and Table 2, the number of decimal places should be consistent.
3. For the line 227-236, there should be more references cited to discuss.
4. Line 265-269, this two paragraphs should be made one paragraph.
5. Line 270-286, authors should strengthen the discussion contents.
6. In the part of 3.3, I’d like to know whether the extraction condition was in dark. Please show us the detail condition.
7. In the part of 3.6, there is no need to show the full name of abbreviation. This problem also happen in the whole part of methods. Please revise it.
8. In the conclusion, it is no need to separate two paragraphs.
Author Response
We would like to thank the editor and reviewers that spent time in this process and add value to the manuscript. The manuscript has been improved according to the suggestions of reviewers; The alterations are represented in the manuscript in red.
----
In this study, five selected DESs were applied for the extraction of bioactive phenolic compounds from avocado peels. The extraction efficiency was evaluated by measuring the total phenolics and flavonoids content. All the results coupled with the non-toxic, biode-gradable, low-cost, and environmentally friendly characteristics of DESs, provide strong evidence that DESs represent an effective alternative to organic solvents for the recovery of phenolic bioactive compounds from agro-industrial wastes. This study is very meaningful to recycle the by-product in food industry. However, there are some critical issues should be clarified as follows:
- The total phenolic compounds contents data should contain standard deviation data in table 1.
Reply: Dear reviewer, all results are expressed as mean ± standard deviation (n = 3).
- For the data in Table 1 and Table 2, the number of decimal places should be consistent.
Reply: Done.
- For the line 227-236, there should be more references cited to discuss.
Reply: Done. New references with results of studies carried out by other authors were added.
- Line 265-269, this two paragraphs should be made one paragraph.
Reply: Done.
- Line 270-286, authors should strengthen the discussion contents.
Reply: The content of the discussion has been changed taking into account the reviewer's comment.
- In the part of 3.3, I’d like to know whether the extraction condition was in dark. Please show us the detail condition.
Reply: This information was included in the manuscript. “duly protected from the light with aluminum foil”. (line 341)
- In the part of 3.6, there is no need to show the full name of abbreviation. This problem also happen in the whole part of methods. Please revise it.
Reply: Done. This problem was corrected in the mentioned section.
- In the conclusion, it is no need to separate two paragraphs.
Reply: Done. The paragraphs were unified.
Reviewer 3 Report
Beatriz Rodríguez-Martínez et al have reported the results of use of Deep Eutectic Solvents as for the Extraction of Bioactive Phenolic Compounds from Avocado Peels. After close evaluation of the manuscript I would suggest revision according to the next points:
1. In Abstract – please provide real results in numbers.
2. In introduction - NADES has been reported for extraction of hydrophilic and lipophilic compounds (https://doi.org/10.3390/molecules26144198) as well as for extraction of phlorotannins (https://doi.org/10.1007/s11094-019-01987-0), phenyletanes and phenylpropanoids (https://doi.org/10.3390/molecules25081826), steroidal saponins https://doi.org/10.3390/molecules26072079 -)..
3. In Sect. 2.1 - How authors may explain, that DES2 and DES5 were more effective for the extraction of flavonoids?
4. In Sect. 2.2 –The phrase “representing 76% of the total concentration of phenolic compounds” is incorrect. “of identified phenolic compounds” ?
5. In Scet. 2.2 – How the composition of DES related to amount individual phenolic compounds content? Please provide HPLC chromatograms.
6. In Sect. 2.3. – Have authors tested FRAP and TAC of pure DES?
7. In Sect. 2.3 – Please discuss the correlation of TPC , TFC and antioxidant activity of extract and update conclusion.
8. Table 3 the MIC are more adequate, than inhibition zone. Which concentrations of DES/DES extracts were used. Please compare concentrations with concentrations of antibiotics.
9. In Sect. 2.4 - Please discuss the correlation of TPC , TFC and antibacterial activity of extract and update conclusion.
10. In Sect 3/1. – Please indicate purity of DES components.
11. Why only acid based solvents were used?
Author Response
We would like to thank the editor and reviewers that spent time in this process and add value to the manuscript. The manuscript has been improved according to the suggestions of reviewers; The alterations are represented in the manuscript in red.
----
Beatriz Rodríguez-Martínez et al have reported the results of use of Deep Eutectic Solvents as for the Extraction of Bioactive Phenolic Compounds from Avocado Peels. After close evaluation of the manuscript I would suggest revision according to the next points:
- In Abstract – please provide real results in numbers.
Reply: Done.
- In introduction - NADES has been reported for extraction of hydrophilic and lipophilic compounds (https://doi.org/10.3390/molecules26144198) as well as for extraction of phlorotannins (https://doi.org/10.1007/s11094-019-01987-0), phenyletanes and phenylpropanoids (https://doi.org/10.3390/molecules25081826), steroidal saponins https://doi.org/10.3390/molecules26072079 -).
Reply: Done. This information has been included in the manuscript. (line 78-83)
- In Sect. 2.1 - How authors may explain, that DES2 and DES5 were more effective for the extraction of flavonoids?
Reply: The composition of these DES turned out to be the best among those studied for the extraction of flavonoid compounds. This was already expected since the great extractive capacity of this type of DES was observed in other raw materials:
De Almeida Pontes, P.V., Ayumi Shiwaku, I., Maximo, G.J., Caldas Batista, E.A., 2021. Choline chloride-based deep eutectic solvents as potential solvent for extraction of phenolic compounds from olive leaves: Extraction optimization and solvent characterization. Food Chem. 352. https://doi.org/10.1016/j.foodchem.2021.129346
Vasyliev, G., Lyudmyla, K., Hladun, K., Skiba, M., Vorobyova, V., 2022. Valorization of tomato pomace: extraction of value-added components by deep eutectic solvents and their application in the formulation of cosmetic emulsions. Biomass Convers. Biorefinery 12, 95–111. https://doi.org/10.1007/s13399-022-02337-z
Saar-Reismaa, P., Koel, M., Tarto, R., Vaher, M., 2022. Extraction of bioactive compounds from Dipsacus fullonum leaves using deep eutectic solvents. J. Chromatogr. A 1677, 463330. https://doi.org/10.1016/j.chroma.2022.463330
- In Sect. 2.2 –The phrase “representing 76% of the total concentration of phenolic compounds” is incorrect. “of identified phenolic compounds” ?
Reply: The change was done.
- In Scet. 2.2 – How the composition of DES related to amount individual phenolic compounds content? Please provide HPLC chromatograms.
Reply: The potential of DES in the extraction of phenolic compounds was proved with the results throughout the manuscript (TPC, TFC, HPLC, Antioxidant Act., ...).
Therefore, the efficiency of DES in the recovery of bioactive phenolic compounds has been proven and compared with an extraction using ethanol as a conventional used solvent.
An example of a chromatogram of the identified compounds at 280 nm from the studied extracts was introduced in a Figure S1 in the supplementary material.
- In Sect. 2.3. – Have authors tested FRAP and TAC of pure DES?
Reply: Yes, the antioxidant activity of pure DES was tested and it was non-existent, so all the activity reported was related to the extracted compounds of avocado peel.
“Our data show that all extracts obtained from avocado peel, regardless of the extrac-tion solvent, have considerable antioxidant activity. Moreover, pure DES showed no anti-oxidant activity for the methods tested.” (line 223)
- In Sect. 2.3 – Please discuss the correlation of TPC, TFC and antioxidant activity of extract and update conclusion.
Reply: Done.
“In other words, the fact that the extractions with the most antioxidant activity are those that have the highest content of phenolic compounds and flavonoids allows us to conclude that it is these compounds that provide the antioxidant capacity to the extracts.”
This was reinforced in the manuscript (line 247 and 428)
- Table 3 the MIC are more adequate, than inhibition zone. Which concentrations of DES/DES extracts were used. Please compare concentrations with concentrations of antibiotics.
Reply: The authors agree with the reviewer's comment. However, this study was carried out with pure DES and the liquid extracts obtained from each extraction (without dilution), so we thought it was good to use the inhibition halo method to perceive the antimicrobial effect.
The antibiotics used in this study were only tested to confirm the non-resistance of bacteria. in this sense, the results of DES and extracts were not compared with antibiotics.
- In Sect. 2.4 - Please discuss the correlation of TPC, TFC and antibacterial activity of extract and update conclusion.
Reply: This was reinforced in the manuscript (line 290-295)
- In Sect 3/1. – Please indicate purity of DES components.
Reply: Done.
- Why only acid based solvents were used?
Reply: Solvents were selected according to previous studies about the same raw material or other similar ones, being the acid-based solvents the most commonly used for this purpose.
Round 2
Reviewer 2 Report
All issues were addressed.
Author Response
Thank you for your comment.
Reviewer 3 Report
Authors have responded to my questions in part and revised the manuscript. However some comments needs additional attention:
1. The phrase "In 235 other words, the fact that the extractions with the most antioxidant activity are those that 236 have the highest content of phenolic and flavonoids compounds allows us to conclude 237 that it is these compounds that provide the antioxidant capacity to the sample." should be supported with relevant reference. Authors have not calculated the correlation coefficient. While the correlation coefficients provide more information (see https://doi.org/10.3390/md20030193).
2. Authors have used lactic aciod of 90#% purity . However in Table 4 no water was indcated for DES1 and DES5. Have authors counted the purity of lactic acid in calculation of DES composition?
Author Response
We would like to thank the editor and reviewers that spent time in this process and add value to the manuscript. The manuscript has been improved according to the suggestions of reviewers; The alterations are represented in the manuscript underlined in yellow.
____
Authors have responded to my questions in part and revised the manuscript. However some comments needs additional attention:
- The phrase "In 235 other words, the fact that the extractions with the most antioxidant activity are those that 236 have the highest content of phenolic and flavonoids compounds allows us to conclude 237 that it is these compounds that provide the antioxidant capacity to the sample." should be supported with relevant reference. Authors have not calculated the correlation coefficient. While the correlation coefficients provide more information (see https://doi.org/10.3390/md20030193).
Reply: The Pearson correlation was performed as suggested by the reviewer. These results were included in the manuscript (table 3 and S1).
2. Authors have used lactic aciod of 90#% purity . However in Table 4 no water was indcated for DES1 and DES5. Have the authors counted the purity of lactic acid in calculation of DES composition?
Reply: The lactic acid used in this study is more than 90% pure. All calculations for the synthesis of DES were made considering the purity of the reagents.